# Nanocomposite Hydrogels as Functional Extracellular Matrices

**DOI:** 10.3390/gels9020153

**Published:** 2023-02-13

**Authors:** Stijn Jooken, Olivier Deschaume, Carmen Bartic

**Affiliations:** Department of Physics and Astronomy, Katholieke Universiteit Leuven, 3001 Leuven, Belgium

**Keywords:** nanocomposite hydrogel, nanoparticles, remote stimulation, biosensing

## Abstract

Over recent years, nano-engineered materials have become an important component of artificial extracellular matrices. On one hand, these materials enable static enhancement of the bulk properties of cell scaffolds, for instance, they can alter mechanical properties or electrical conductivity, in order to better mimic the in vivo cell environment. Yet, many nanomaterials also exhibit dynamic, remotely tunable optical, electrical, magnetic, or acoustic properties, and therefore, can be used to non-invasively deliver localized, dynamic stimuli to cells cultured in artificial ECMs in three dimensions. Vice versa, the same, functional nanomaterials, can also report changing environmental conditions—whether or not, as a result of a dynamically applied stimulus—and as such provide means for wireless, long-term monitoring of the cell status inside the culture. In this review article, we present an overview of the technological advances regarding the incorporation of functional nanomaterials in artificial extracellular matrices, highlighting both passive and dynamically tunable nano-engineered components.

## 1. Introduction: Nanocomposite Hydrogels for Tissue Engineering

In living tissues, cells are supported by a three-dimensional (3D) extracellular matrix (ECM), a dynamic structure, highly optimized for triggering and supporting cell functions such as adhesion, differentiation, migration, and polarization. The ECM is what glues together the cells and organs and is composed of a variety of proteins, ranging from large glycoproteins and proteoglycans to collagens, displaying various biochemical domains for specifically interacting with cell membrane receptors and featuring heterogeneous physical properties [1,2]. The ECM typically takes shape as networks of fibers with dimensions ranging from nano- to micrometers, carrying a diverse range of (bio-)chemical and (bio-)physical cues to regulate cell behavior. Biochemical cues comprise adhesion ligands, chemical functional groups (determining fiber hydrophilicity, charge, protein adsorption, etc.), and soluble factors [3,4,5]. Biophysical cues, on the other hand, encompass hydrogel mechanical properties, orientational anisotropy, biodegradability, and electrical conductivity [6,7].

In addition to static biophysical ECM properties, in vivo, cells also experience local and dynamic physical gradients, including strain [8], stress [9,10], electric [11,12], and thermal fields [13,14]. Significant effort has for instance been dedicated to understanding mechanotransduction mechanisms [7], developing strategies to control tissue development and repair by tuning mechanical properties of essentially isotropic materials (mostly topography and stiffness), and eventually applying external stimuli to the bulk macro-environment such as shear stress [15] or electric current [16,17]. The research efforts to understand and recapitulate the specificities of natural ECMs and eventually engineer functional synthetic tissues have shown that cells display different phenotypes and genotypes in in vitro two-dimensional (2D) monolayer ‘Petri dish’-like cultures compared to cells suspended in a 3D hydrogel consisting of ECM proteins [18,19,20]. Research hence aims at mimicking the natural cell environment through designing hydrogels of ECM proteins for reproducing in vivo cell behavior [3]. In recent years, a large variety of engineered (nano)materials has been produced and evaluated for their ability to promote tissue development and regeneration [21,22]. Various artificial 3D ECM and 2D cell culture materials have been developed to replicate different tissue-specific mechanical properties, accommodate chemical binding motifs or dynamically release small molecules triggering specific cellular responses. Nano fibers have been engineered to mimic the properties of protein fibrils and fibers present in natural ECMs, such as collagen [23,24,25], fibrin [26,27], elastin [28,29], possibly in combination with adhesion proteins (laminin, fibronectin) [30,31] or glycosaminoglycans (GAG) including heparin, keratin, etc. [32] Synthetic hydrogels (e.g., polyethylene glycol (PEG), polyacrylamide (PA)) have been developed as well, allowing for a plethora of chemical modifications [33]. All of these have their respective benefits and drawbacks that should be considered when a hydrogel is selected. Some natural protein fibers like collagen or fibrin, for instance, are often used without any modification, while synthetic materials like PEG require some form of chemical modification to support hydrogel crosslinking and cell adhesion. Limitations of collagen and fibrin, among other natural materials, include typically low stiffness values, limited long-term stability, and, especially, high batch-to-batch variability. Synthetic PA or PEG polymers allow better control of mechanical properties. PA crosslinking, however, involves possibly cytotoxic precursors and is less well-suited for cell encapsulation, while PEG is more cytocompatible. There are several reviews in the literature addressing a huge spectrum of (fibrillar) nanomaterials, their properties, and their synthesis routes together with the obtained cellular responses [34,35,36].

To emulate the dynamic properties of ECMs, on the other hand, new synthetic cellular scaffolds combining the cellular affinity of biological nanofibers with the functional properties of inorganic NPs have emerged over the past years. The field of nanocomposite hydrogels was inspired by the initial development of nano-reinforced bone tissues [37] containing mineral nanoparticles such as calcium phosphates (e.g., hydroxyapatite), silica [38], and silicate [39], where it was first shown that the presence of nanocrystallites enhances osteoblast mineralization through an increased scaffold toughness [40]. To date, the field has evolved, and various types of engineered nanomaterials have been incorporated into TE scaffolds to better mimic the conditions of the in vivo cell micro-environment. Nanoparticles (NPs) can potentially provide a technological solution for both remote, localized stimulation at the cellular level and remote read-out of the actual state and development of the cells in artificial tissues. Regarding the former, NPs can respond to remote external stimuli, e.g., light, magnetic fields, etc., and, as such, allow for dynamically delivering global or localized physical stimulation to the cells, e.g., through the generation of E-Fields and temperature gradients, modulating cell functions. On the other hand, NPs can be used as sensors for monitoring the status of a developing tissue construct, e.g., through observation of pH, temperature, secreted molecules, metabolites, oxygen levels, etc. Especially, optical methods are promising as NPs are very sensitive to their environment, often alleviating the need for biorecognition elements such as antibodies. Figure 1 gives an overview of the cues incorporated in nanocomposite hydrogels for TE, distinguishing static biochemical and physical cues as well as functional nanocomponents for static reinforcement and dynamic tunability.

The scope of this review is limited to the application of nanomaterials in combination with ECM materials for synthetic tissue constructs, in particular fibrillar hydrogels. In the following sections we discuss hybrid hydrogel-nanomaterial scaffolds, where the nanoparticles influence mechanical properties to modulate cell differentiation, migration, proliferation, adhesion, etc.; improve electrical activity, provide localized stimuli, and read out the resulting cellular responses. In Section 2, we provide an overview of research targeting the enhancement of one of the traits of artificial ECMs using NPs, including mechanical properties, electrical conductivity, and anisotropy of the scaffolds, as well as to control cell release from the scaffold. Section 3 and Section 4 are dedicated to the use of NPs as transducers of external stimuli into localized, cellular cues, and biosensing, respectively. Other biomedical applications of nanoparticle-hydrogel superstructures, e.g., drug-delivery vesicles, biosensors and -actuators, biodetoxification [41], and immune modulation [42] to name a few, are beyond the scope of this review. Readers interested in such applications are referred to other publications [43,44,45].

## 2. Nano-Enhanced Tissue Scaffolds

### 2.1. Nanoparticles for Mechanical Reinforcement

In the absence of nano-additives, the bulk mechanical properties of hydrogel-based artificial ECMs are commonly tuned by varying the polymer concentration or the amount of cross-linker [46,47]. Highly cross-linked, small mesh size hydrogels can however limit cell proliferation and migration and, as such, inhibit the formation of functional tissue constructs. As an alternative, the incorporation of nanoparticles in polymer networks allows for effectively transferring the mechanical properties of the particles to the scaffold, given sufficient particle-polymer interaction, tuning its bulk mechanical properties without increasing gel density or reducing pore size. Especially when the nanofillers are incorporated as crosslinkers, the mechanical properties, such as Young’s modulus, stiffness, fracture and fatigue resistance, strain at break, etc., can be considerably higher compared to non-covalent interactions [48]. Most commonly, bone and cartilage TE benefit from enhanced mechanical properties, but also skeletal and cardiac constructs have been developed, employing conductive NPs to simultaneously enhance the scaffold’s mechanical properties [37,49] and/or electrical conductivity [50,51]. The latter is discussed in more depth in Section 2.2.

A wide range of NPs have been employed for scaffold mechanical property optimization, ranging from polymeric NPs to hydroxyapatite [52], calcium phosphate [53], nanoclay [54], silica [55], bioglass [56], graphene (oxide) [57], carbon nanotubes (CNT) [58], silver [59], titanium [60], gold nanoparticles (GNPs) [61] or iron oxide [62]. For example, the inclusion of 0.5 mg/mL CNTs in 5% methacrylated gelatin hydrogels led to a threefold increase in the tensile modulus [63], gold nanoparticle (GNP) incorporation into poly(N-isopropylacrylamide) (PNIPAM) even resulted in a sixfold enhancement of the shear modulus [64]. CNTs and GNPs have been encapsulated at concentrations ranging from 0.5 mg/mL up to 5 mg/mL [58,61,63] to increase scaffold mechanical properties in combination as well as electronic conductivity, while Fe_3_O_4_ and TiO_2_ have been incorporated into soft matrices at concentrations up to 5% [60,62]. Based on the reported cell viability studies these relatively high NP concentrations did not cause any significant cell toxicity over a time span of 7 days. In another work, laponite (clay) infused gelatin-methacrylate (GelMA) displayed a fourfold increase in Young’s modulus and was shown to induce bone mineralization even in the absence of osteo-inductive growth factors [52,54]. Similarly, increased hydrophilicity and elastic modulus of poly(lactic-co-glycolic acid)/polycaprolactone (PLGA/PCL) electrospun fibers containing iron oxide NPs also positively influenced osteogenic differentiation of the stem cells cultured on top [65]. In recent work, Arno et al., specifically studied the effect of the morphology of polymeric poly(L-lactide) based NPs (PLLA) on the bulk mechanical properties of alginate gels [66]. They showed the possibility of controllably tuning the mechanical properties through the morphology and concentration of the nanofiller. A nanocomposite containing platelet shaped PLLA NPs, for instance, showed enhanced resistance to breaking under strain compared to their spherical and cylindrical nanofiller counterparts. Upon gelation in the presence of spherical NPs, strain-dependent oscillatory rheology showed a small increase in strain at the flow point (crossover point of G′ and G″) such that the materials flowed at ~25% strain at break, compared to ~10% when no additive was used, and 15% for cylindrically-shaped PLLA NPs, the latter which is not significantly different from gels without NP additive. Moreover, platelet-infused hydrogels also showed an increasing strain at the flow point demonstrating those gels can withstand higher shear rates before breaking. Apart from particle geometry and concentration, the viscoelastic properties of nanocomposite hydrogels also depend on the interaction between NP and fiber components [48,67].

### 2.2. Nanomaterials Controlling Structural Alignment

The ‘conventional’ nanocomposite hydrogel systems, however, lack the mechanical anisotropy inherent to the long-range order typically found in in vivo ECMs. Tissues are characterized by a highly anisotropic, hierarchically ordered ECM architecture. The isotropic nature of randomly oriented polymer networks in 3D hydrogel systems limits the ability to mimic the complex hierarchical anisotropy in native ECMs. Modification of the hydrogel microstructure is needed to obtain anisotropic organization and drive tissue organization [68]. Symmetry breaking cues, in the ECM mechanical properties, are essential for multiple cell types such as neurons and cardiomyocytes [69,70]. Symmetry breaking is for instance deemed responsible for the polarization of the contractile cytoskeletal organization in cardiac and skeletal muscle cells [71] as well as the formation of axons and the assembly of functional synapses in neural networks [72]. Current methods for scaffold alignment such as shear force [73], unidirectional compression [74], or freeze casting [75] have yet to provide the desired control over the architecture of the hydrogel. The incorporation of aligned nanofillers, on the other hand, can induce mechanical anisotropy into a polymer hydrogel, and guide the directional growth of the encapsulated cells. In this scenario, the nanomaterial is typically incorporated into the hydrogel precursor solution or grafted onto the polymer matrix by modifying the NPs with crosslinkers [76,77], subsequently aligned either by applying shear stress to the hydrogel (e.g., bioprinting) [78], or external magnetic or electric fields and, finally, fixated in place by hydrogel curing [79].

Electric fields can orient nanofillers such that their electric dipole moment is aligned with the externally applied E-Field. For instance, dielectrophoresis has been used to align CNTs in gelatin methacrylate gels used for the 2D culture of muscle cells [80]. The resulting alignment was shown to stimulate skeletal muscle cells in the production of more functional myofibers [81] as well as enhance cardiac differentiation of mouse embryoid bodies when compared with pure GelMA scaffolds [82]. In scaffolds containing metal or semiconductor NPs, the electrogenic cells arguably benefit from electric conductivity as well as mechanical anisotropy. Other examples of electrically oriented nanostructures include clay nanosheets [83], silver nanowires [84], both in PNIPAM matrices, silk nanofibers [85], and barium titanate NPs in gelatin [86]. The latter study demonstrated the increased elastic modulus of the resulting piezoelectric hydrogel under the application of an external E-Field. However, all these studies were limited to 2D cell cultures onto the anisotropic/conductive scaffolds, since alignment requires relatively strong electric fields (2 kV/cm in Refs. [80,81,82], 1.25 kV/cm in Ref. [84] and 27 kV/cm in Ref. [86]) which would influence or even damage embedded cells in 3D cultures.

The most investigated area in this field is alignment driven by magnetic nanofillers [87]. In these studies, the magnetic material of choice is mainly superparamagnetic iron oxide, although other (super)paramagnetic, ferromagnetic, and occasionally even diamagnetic materials have been employed as well. Superparamagnetism, appearing in small (i.e., tens of nm), single-domain, ferro- or ferrimagnetic (e.g., magnetite Fe_3_O_4_) materials, is required to precisely control NP behavior and avoid aggregation inside the so-called ferrogel. As a matter of fact, super paramagnetic iron oxide NPs are FDA-approved (U.S. Food and Drug Administration) and are already present in several commercial contrast agents for magnetic resonance imaging (MRI), and used in medical diagnostics, hyperthermia-based therapy cancer treatments [88], or drug delivery applications. Ferro- or (super-)paramagnetic fillers, like iron oxide [89,90], nickel nanorods [91], barium ferrite, and carbonyl iron NPs [92,93,94], can be oriented with field strengths of less than 1 T. To orient diamagnetic materials, superconducting magnets are required to create fields larger than 5T. In this category, (functionalized) CNT [93,95], (reduced) graphene oxide ((r)GO) [96], inorganic nanosheets, and cellulose nanocrystals [97,98,99,100] have been used. Moreover, some fiber materials such as collagen [101] and fibrin [102] are diamagnetic, and can also be aligned at high field strengths (up to 8 T in [101,102]).

Cell culture within or on top of mechanically anisotropic matrices generally results in the alignment of the cells along the directional vector of the nanostructures. Tognato et al., for instance aligned iron oxide NPs (IONPs) into filaments of tunable sizes within a GelMA scaffold through the application of a low-intensity magnetic field. C2C12 skeletal myoblasts seeded on top or embedded in the nanocomposite hydrogel, aligned along the axes of the IONP filaments. Furthermore, in 3D, the myoblasts differentiated toward myotubes even in the absence of differentiation media [103]. In an alternative approach, magnetic NPs have been used to align synthetic or protein fibers within the hydrogel. Betsch et al. incorporated streptavidin-coated IONPs in printable bio-inks containing various concentrations of low-temperature gelling agarose and type I collagen. By applying a magnetic field during bioprinting, movement of the IONPs through the hydrogel resulted in aligned collagen fibers within hydrogels containing 0.2% collagen in 0.5 (*w*/*v*)% agarose [104]. Similarly, a fibrin-based hydrogel containing aligned superparamagnetic IONP-doped, electrospun PLGA microfibers, stimulated fibroblasts, and nerve cells into linearly directional growth along the direction of the microfibers [105]. In fact, even minimal structural guidance was proven to affect neuronal alignment in 3D anisotropic hydrogels [106]. 3D encapsulated fibroblasts were shown to respond to the incorporation of 1.5% of PEG microgels, magnetically aligned through the incorporation of rod-like superparamagnetic IONP (0.0046%) within a fibrin hydrogel. The incorporation of 1.0% magnetic PEG microgels proved sufficient to induce directional growth of a 3D culture of embryonic chicken dorsal root ganglion (DRG) neurites (see Figure 2C).

Thus, nanomaterials bearing magnetic properties can be used to assist the preparation of complex anisotropic TE scaffolds, and potentially augment their functionality. The nanomaterial used in conjunction with a magnetic field may hold the cells into place, with or without a crosslinked fiber network. Upon release of the magnetic field, the magnetic support loosens, eventually releasing entrapped substances and cells [107]. This approach offers therefore a high degree of spatial and temporal control over the scaffold positioning and functionality [108]. Moreover, magnetically actuated nanoparticles may also be used to modify and guide cells to desired locations, assisting, in turn, the formation of cell-containing scaffolds and cell sheets. This approach was successfully employed to prepare implantable cell sheets for reparative angiogenesis from dilute ECM precursor (collagen, Matrigel), and liposome-encapsulated nanoparticles [109], or for dental pulp regeneration, by loading GO-magnetite hybrid NPs in dental pulp stem cells and guiding them to form multilayered cell sheets of controlled geometries using a magnetic field [110]. Human aortic smooth muscle cells modified with magnetically tagged bacterial cellulose were used by Arias et al., to concentrate cells and stimulate the formation of vascular grafts at desired locations [107]. The cell sheet approach and other, so-called magnetic TE approaches are especially useful to assemble cells into complex geometries. Stable multilayer keratinocyte constructs can, for example, be prepared from cells loaded with magnetic particles, while keratinocytes do not produce enough ECM proteins to stabilize such assemblies under classical cell culture conditions [111].

In addition to assisting the production of synthetic tissues, magnetic particles have also been employed to dynamically remodel cellular matrices by altering the bulk mechanical properties or orientational hierarchy [112]. Temporal control over the mechanical properties of the cell microenvironment is of tremendous importance in the study of cell mechanotransduction. It has been reported, for instance, that stem cells retain a history of experienced mechanical cues, affecting behavior at a later stage in their development [113,114]. Manipulation of carbonyl iron NPs in a carrageen scaffold allowed a dynamic and reversible stiffening of the matrix, stimulating the secretion of proangiogenic molecules, and dynamically controlling osteogenesis [115,116], or, in another work, regulating the adhesion of endothelial cells [117].

### 2.3. Nanoparticles Enhancing Electrical Conductivity

The inclusion of conductive NPs, as briefly touched upon in the previous section, also enhances the electric conductivity of the hybrid scaffolds [118,119]. This is very relevant for engineering electrically active tissues such as cardiac and skeletal muscle or nervous tissue [50,120,121]. The native myocardium, for instance, has a conductivity of 0.1 S/cm and can establish synchronized contraction due to direct electrical coupling between cells through gap junctions [63]. Reduction of the electrical impedance of a cardiac tissue scaffold is hence crucial for accurately mimicking the native cardiomyocyte microenvironment and engineering functional cardiac patches. As such it has been shown for a wide variety of carbon- (e.g., polyaniline [122], CNT [82,123,124], GO [125]), metal-based NPs (e.g., GNP [126,127,128], silver [129], iron oxide [130]), among others [131], that NP incorporation resulted in enhanced matrix conductivity, improving the electrical signal transfer between cardiac cells [132], increasing cell alignment, elongation, striation as well as promoting the expression of cardiac markers such as Troponin I and Connexin-43 [132]. Lee et al. (see Figure 3) showed that conductive CNT- (100 kΩ/sq sheet resistance) and rGO-GelMA (1 kΩ/sq sheet resistance) scaffolds significantly improve the expression of functional protein markers in a top culture of cardiomyocytes, compared to their non-conductive GO-GelMA scaffolds [133]. Electrophysiology recordings revealed that the obtained cardiac tissue constructs displayed distinct cardiomyocyte phenotypes and different levels of maturity based on the nanocomposite substrate, more ventricular-like on CNT-GelMA, atrial-like on GO-GelMA, and a mixture of both on rGO-GelMA. In another work, gold nanowire-alginate composite scaffolds featured thicker, more aligned cardiac tissues compared to traditional pristine scaffolds [134]. Under bulk, external electrical stimulation, cardiac cultures on top of electrically conductive scaffolds displayed more uniform, synchronized contractions throughout the entire cardiac tissue with lower excitation thresholds [134]. Specifically, electrical stimulation of a GNP-reinforced conductive matrix allowed to generate uniform synchronous contraction at frequencies from 0.5 to 2 Hz, which was not possible on GNP-free scaffolds, using a relatively low stimulation threshold of less than 4 V [17,127,135,136,137]. Moreover, in another study, electrical pulse stimulation showed an enhanced differentiation of stem cells on a CNT-reinforced hybrid scaffold [82].

In combination with NP alignment, electrical anisotropy can also be induced [80,138]. Aligned CNTs [81,82,138], or graphene [139] promoted a higher conductivity of the scaffold compared to their randomly distributed counterparts. Myofibers demonstrated more maturity and contractability after stimulation along the alignment direction [80]. Moreover, cardiac cultures on top of aligned CNT forest microelectrodes embedded in GelMA and PEG hydrogels showed improved cell-to-cell coupling and maturation. They displayed well-defined, elongated, and interconnected sarcomeric *α*-actinin structures, as well as a homogeneous distribution of Connexin-43 with enhanced synchronous contractile properties throughout the entire cardiac tissue construct. The excitation threshold during electrical actuation was highly dependent on the direction of the electric field, up to 5 times lower along the alignment direction compared to perpendicular stimulation [138].

Electro-active nanocomposite scaffolds for the study of neuronal behavior, on the other hand, have been shown to result in an enhanced attachment of Schwann cells [140], boosted proliferation and spreading of PC12 cells concomitant with enhanced differentiation, boosted neurite outgrowth upon BNDF (Brain-derived Neurotrophic Factor) exposure [141,142,143] and improved neurite outgrowth in rat primary neurons [143]. Specifically, the incorporation of CNTs in a collagen hydrogel enhanced neurite outgrowth of rat neonatal dorsal root ganglia. Upon electrical stimulation, a 7-fold increase was observed in comparison with stimulated bare collagen scaffolds [144]. Notably, most nanocomposite scaffolds for nerve regeneration are combined with anisotropy creating nerve conduits [142,145] or aligned scaffolds [146,147] for directional growth.

To conclude this section, nano building blocks have been added to hydrogels initially for enhancing or adding tissue specific physico-chemical properties to hydrogel matrices supporting cell differentiation and maintenance, for example, to achieve higher mechanical moduli in materials emulating load-bearing tissues (i.e., bone or cartilage), or electrical conductivity for electrogenic tissues (e.g., cardiac or nerve tissues). More recently, nanoparticles in combination with electrical and magnetic fields allowed to induce structural anisotropy in the bulk of a cell scaffold. The diversity in nanomaterial physical properties and their sensitivity to external stimuli allows for rendering various functionalities to hydrogel materials while using the smallest possible quantity of material. On the other hand, the added value should be always considered against biocompatibility and the potential long-term effects of nano-additives on tissue functionality. So far, most studies have only evaluated short-term effects (from a few days to a few weeks) on cell viability and development, thus, data on cell behavior after long term exposure is still to be generated. Moreover, controlling the nanomaterial distribution in the final hydrogel is not trivial, as it is achieving an optimal balance between the physical properties of the scaffold and low cell toxicity.

## 3. Tunable, Remote, and Localized Cell Function Modulation

In vivo, the chemical and physical cues in the ECM exist in a dynamic state, spatially and temporally, regulated by cell functions. Rather than merely mimicking the ECM static morphological features, NPs exhibiting plasmonic, magnetic, and/or optical properties could allow for dynamically changing the matrix properties, delivering local stimuli (e.g., temperature, electric fields, mechanical stress, drug release, etc.) within the scaffold or they could sense in situ cellular responses (e.g., electrical activity, acidification, the release of inflammatory markers, etc.). In these scenarios, the NPs incorporated into the nanocomposite scaffold should be able to convert external wireless stimuli into a secondary signal, modulating cell functions, or vice versa, a cellular event should cause a detectable change in the physical properties of such NPs. While the former would allow to dynamically guide cell behavior, the latter functionality would enable the non-invasive monitoring of the cellular responses, thus in situ reporting the status of the tissue construct. In this section, nanomaterials that impart local, remotely triggered stimulation, to which the tissue responds or detection functionalities, revealing important information on the proper functioning of the engineered tissue, are discussed.

### 3.1. Thermal Stimulation

Through the opto-thermal or magneto-thermal [148,149,150] effect, NPs convert an external stimulus, light, and magnetic field, respectively, into heat. Cells are highly sensitive to temperature changes, adjusting their metabolism and growth patterns accordingly. Hence, (localized) thermal stimuli can be used to tune cell adhesion [151], migration [152], differentiation, and even electrical activity [149,153]. Most developed nanocomposite scaffolds capable of generating localized heat are photo-responsive materials containing plasmonic NPs.

GNPs are the most widely used optical absorbers converting near-infrared (NIR) light into heat via localized surface plasmon generation and are often chosen over other metallic nanostructures (Ag, Pt), carbon particles, and other potential mediators [154,155,156,157,158,159,160] because of their biocompatibility and tunable plasmon absorption bands. For details regarding other photothermal nanoparticles that have recently been developed for biomedical applications, readers are invited to consult recent reviews specialized in the topic [161].

GNP sizes and/or shapes can be tuned such that their absorption bands match the NIR biological transmission window of living tissues. So far, the best tunability and optical characteristics were obtained with gold nanorods (GNRs) and gold nanoshells (GNSs) [162]. In fact, GNP-based therapies are in clinical testing for hyperthermia cancer therapies and are already used for various other treatments [163].

Plasmonic heating of GNPs has been shown to enhance neurite length in HG108-15 neuronal cells [164,165], induce myotube contraction in striated muscle cells [166,167] or when targeted to ion channels in the cell membrane, trigger membrane depolarization of dorsal root ganglion neurons and mouse hippocampal as well as trigeminal neurons resulting in AP firing [168,169]. Moreover, in vitro, APs have been induced in NIR stimulated rat auditory neurons incubated with silica-coated gold nanorods [170], and in vivo, sciatic nerves injected with GNRs were six times more active in firing compound APs, with a threshold three times lower than control [171]. For long-term stimulation strategies, however, the internalization of the GNPs, proven difficult to control, is causing variable and sometimes inconsistent results. As opposed to the above studies where neuronal firing is stimulated, Yoo et al., reported a reversible inhibition of neural activity by GNRs in combination with optical stimulation. In this study, GNRs were electrostatically bound to the cell membrane of hippocampal, cortical, and olfactory bulb neurons, and their NIR illumination was reversibly inhibiting electrical activity. Temperature-sensitive inhibitory TREK-1 channels were deemed responsible for the observed effect [172,173]. Experimental evidence suggests that local heating can cause changes in the membrane capacitance [174,175]. Specifically, it was proposed that TRPV ion channels are activated by heating, leading to depolarization, which finally activates sodium channels and generates an AP [176]. The contrasting results obtained in different studies demonstrate that the understanding and control of GNPs-based mediation of cellular activity is still in its infancy. The lack of control on the positioning of the NPs with respect to the cell membrane and the multiple ways in which the stimulated NP can influence membrane voltages (i.e., plasmon generation can induce not only local heat but also strong electric fields, charge transfer, etc. and can thus activate different types of cell receptors simultaneously) may all play a role in the widely variable cell responses observed.

Besides the activation of electrogenic tissues, several studies have demonstrated cell migration guidance or cell sorting using heat. In 2D cultures, Min Zhu et al., fabricated RGD-coated (arginylglycylaspartic acid) gold nanoarrays that influenced integrin-mediated adhesion by functioning as plasmonic heaters [152]. The light-induced generation of thermal gradients allowed to guide and block the migration of mouse fibroblasts. Especially the activity of cells involved in the body’s immune response is regulated by mild, fever-like variations in temperature. The migration kinetics and expression of activity markers such as CD86 of epidermal dendritic cells have for instance been shown to be enhanced by mild heating [177,178]. In 3D hydrogel systems, dynamic cell guidance has been influenced as well by mild, localized heating in combination with temperature-induced variations in hydrogel mechanical properties through either shrinking/swelling, phase changes, or even gel denaturation [151,179,180,181]. Hribar et al., for instance, formed channels into the matrix by collagen denaturation through NIR heating of the embedded GNPs. The channels allowed the cells to migrate, proliferate, and align [180]. In this scenario, the dynamic response of the cells is rather governed by the change in scaffold structure than by the temperature-induced cell responses. Such approaches are discussed in more detail in Section 3.3 on mechanotransduction.

In addition to the challenge posed by the precise positioning of nanoactuators with respect to cellular membranes and receptors, leading to a wide spectrum of cellular responses, another current limitation of optothermal methods for in vivo and in vitro applications relates to the scattering and absorption of electromagnetic waves by living tissues. Although multiphoton techniques enable increasing both the spatial resolution and depth at which tissue can be stimulated, magnetic fields combined with magnetic nanoparticles have recently shown high potential for reaching deep body regions, for example in the context of deep brain stimulation [182]. More specifically, untargeted, superparamagnetic iron oxide NPs, for instance, were used to induce heat-generated Ca^2+^ influx in TRPV1-expressing (transient receptor potential vanilloid) Hek T293 cells [149,153], to stimulate dissociated hippocampal neurons expressing TRPV1 to fire action potentials (APs) [149,153] and activated neurons at the ventral tegmental area of mice to enhance the expression of c-fos upon the application of an alternating magnetic field [149]. Initially limited in spatial resolution with respect to optical stimulation techniques, the combination of refined magnetic actuators and particle targeting to specific tissues and cell types will no doubt provide an excellent alternative to photothermal stimulation for deep regions of native or artificial tissues.

### 3.2. Electric Stimulation

The membrane potential of (electrogenic) cells can be modulated by localized external electric fields, activating voltage-gated ion channels, changing the capacitance of the cell membrane and/or its porosity or tension [156,183,184,185]. Delivery of a nano-mediated electric stimulus can be achieved through a light, a magnetic or an acoustic pulse, employing NPs with opto-, magneto-, or acousto-electric properties, respectively.

Opto-electric effects in semiconductor quantum dots (QDs) were reported to stimulate neuronal cells in vitro. When optically excited, the electric dipole moments associated with the generated exciton create local electric fields sufficiently large to activate voltage-gated ion channels. Cell electrical activity in response to QD excitation was reported by Pappas et al., in neuroblastoma NG108 cells cultured on top of poly(dimethyldiallylammonium chloride)—HgTe thioglycolic acid stabilized QD multilayers in combination with a polylysine/poly(acrylic acid)/polylysine multilayer or clay sheets for better biocompatibility [186]. Upon optical stimulation, cells voltage-clamped at −65 mV showed an instantaneous inward current matching the time course of the stimulation. Using clay sheets for the layer-by-layer films, Pappas et al., were capable of eliciting action potentials in only about 11% of the cells, suggesting the need for a more efficient and biocompatible interface. 2D cultures of cells on top of QD films were further shown to be able to elicit action potentials in cortical neurons cultured on top of a CdSe QD film [187] (see Figure 4), PC12 and Neuro2A cells cultured on photoelectrodes coated with InP/ZnO QDs [188]. Finally, CdSe/CdS core-shell nanorods conjugated onto CNTs could stimulate a chick retina that lacked functional photoreceptors. In this study, the film maintained its stability and biocompatibility for 21 days [189].

Using a similar methodology, Guduru et al., used CoFe_2_O_4_·BaTiO_3_ paramagnetic core-shell NPs to modulate deep brain circuits through magnetoelectric stimulation [190], while Marino et al., employed acousto-electric transduction to induce Ca^2+^ influx in SH-SY5Y cells in the vicinity of BaTiO_3_ NPs [191].

All these studies were performed on 2D cell cultures on top of QD-coated substrates and emphasized the importance of NP positioning with respect to the cell membrane for efficient signal transfer. This is because although a sizable electric field exits over the lipid bilayer, it rapidly dissipates in the extra- and intra-cellular domain due to screening by polarized water and membrane-adjacent ions. The distance over which it exponentially decays is characterized by the Debye length, representing the length scale over which the electric field drops to 1/e of its initial value and typically measures roughly 1 nm. Therefore, to modulate the transmembrane potential, precise NP positioning is crucial, and nm shifts in particle position can determine experimental success or failure [192,193].

NP coupling onto ECM structural components might in principle improve the positioning and stability of the functional NPs. Dhandayuthapani et al., for instance, constructed a hybrid QD-fiber scaffold consisting of Zein electrospun fibers containing CdS QDs aimed at various applications, including electro-active scaffolds for wound healing, fluorescent visualization of scaffold structure, and electrochemical sensing. It was successfully tested for the culture of mesenchymal stem cells and fibroblasts, showing no negative impact on cell viability [194].

Similarly, Wei et al., prepared fibronectin nanofibers modified with CdSe/ZnS QDs [195]. These QD-functionalized nanofibers could be used as building blocks for future photonic, smart functional cell scaffolds. The nanofiber matrices could be used as such or ligand modified to promote cellular adhesion and provide the required mechanical and morphological cues to promote cell viability, allowing a robust NP anchoring close to the cell membrane for optimal signal coupling, while limiting NP endocytosis.

### 3.3. Nanoscale Mechanical Properties and Mechanotransduction

The cellular native micro-environment is comprised of fluctuating endogenous and exogenous forces at multiple length scales. Endogenous forces are generated through the interaction of a cell with the extracellular matrix and transmitted through focal adhesion sites. Neighboring cells sense these forces as they are conveyed to their internal machinery and respond through adhesion, deformation, and migration, applying forces similar to the ones they respond to [196,197]. This process is called mechanotransduction. Additionally, exogenous forces such as shear flow or gravity influence numerous cellular processes [7]. These mechanical forces have a significant impact on gene expression and regulation of tissue functions. Understanding the translation of mechanical forces into different biological outcomes is crucial for many applications from developmental biology to regenerative medicine and cancer research. Yet, it is limited by the availability of platforms for allowing force delivery to cells and their extracellular environment, at different length scales, and single cell level, particularly in 3D cultures. In this section, we will discuss active materials interesting for mechanotransduction studies, i.e., materials capable of delivering localized mechanical stimuli to cells and/or inducing dynamic matrix remodeling (e.g., a change in mechanical modulus) through remote methods.

Essentially, many hydrogel systems are inherently sensitive to environmental factors, such as pH [198,199], light [114,200,201], temperature (e.g., PNIPAM) [202,203], etc., by translating a change in the environmental conditions into a specific configuration of the polymer network, for instance causing swelling/shrinking upon change of pH or temperature. This feature makes hydrogels excellent transducer elements to translate an external stimulus into mechanical deformation. Yoshikawa et al., for instance, used the pH-sensitive poly(2-(dimethylamino)ethyl methacrylate) (PDMAEMA) as a building block for a hydrogel which reversibly changes its conformation upon small changes in pH, causing hydrogel swelling/shrinking. Alternatively, light-responsive hydrogels often rely on photo-reversible cross-linking [204,205] or thermoplasmonic NPs [202]. As such, Wu et al., used the photo-activated protein Dronpa as a crosslinker to reversibly tune the mechanical properties of a PEG hydrogel [205]. To read out the magnitude of the applied stimulus, stimuli-responsive hydrogels have been combined with nano- or micro-structured arrays, where the bending/tilting of the flexible posts reveals the magnitude of the external stimulus [206,207]. Sutton et al., used this approach to deliver µN forces to D1 ORL UVA cells cultured on top of a PNIPAM hydrogel (see Figure 5) [207]. Through the addition of GNP plasmonic heaters, a localized light stimulus was transformed into a highly localized, directional, and reversible matrix deformation, of which the magnitude was read out through bending of an array of 2 µm × 10 µm × 12 µm microposts fabricated using Norland optical adhesive. Similarly, NIR irradiation of a CNT-reinforced collagen-PNIPAM gel could locally and reversibly transmit mechanical forces (up to 5% strain) to the human fetal hepatocytes cultured on top [208]. Through collapsing the thermo-responsive poly(N-isopropylmethacrylamide) (PNIPMAM) surrounding plasmonic GNPs, pN forces can be applied with a subcellular resolution to specific cell receptors, controlling integrin-based focal adhesion, cell protrusion, migration, and T-cell receptor activation [209,210].

In our lab, the concept was extended from 2D to 3D cell cultures by designing a GNR-containing PEG hydrogel, as illustrated in Figure 5A, able to generate heat and deformation through NIR light control [211]. The scaffold was combined with optically addressable temperature probes (more details in Section 4). To illustrate the localized mechanical actuation, encapsulated SH-SY5Y cells were periodically stimulated into a ‘beating’ motion, known to elicit various cellular responses [200], through the deformation of the PEG matrix under periodic NIR illumination.

Another application requiring dynamic control over the scaffold’s mechanical properties is selective cell detachment (or adhesion) for cell sheet engineering and cell sorting. Kojima et al., developed an approach to selectively detach single cells by adding GNPs to a collagen gel. By irradiating the gel with green light, the generated heat locally caused the denaturation of the surrounding collagen into gelatin, thereby releasing the cell cultured on top [212]. HeLa, colon-26, MCDK epithelial cells, and SH-SY5Y could be removed with a close to 100% success rate [151,212]. Such cell-sorting methods may provide a valuable addition to fluorescence-activated cell-sorting methods by enabling selective harvesting of the target cells, both in space and time, from a cell culture based on cell morphological traits as well as fluorescent labels. Moreover, the use of NIR light combined with thermoplasmonic materials is milder for cells compared to the use of UV-photolabile materials, since UV radiation leads to higher phototoxicity and less depth penetration (particularly relevant for 3D constructs). Similarly, Cui et al., grafted PNIPAM onto silicon nanowires. NIR photothermal heating of the nanowires causes reversible phase transformation of PNIPAM from a hydrophilic to a hydrophobic state, allowing for cells to adhere at 37 °C and detach at a temperature below the phase-transition point [213].

In culture systems where nanoscale mechanical actuation is generated through the heating of plasmonic nanoparticles, it is difficult to uncouple the effect of temperature and mechanotransduction on cellular behavior. The heating of thermoplasmonic GNP arrays was also shown to controllably drive cell migration in the absence of mechanical stimuli [152]. Alternatively, electric fields can mechanically deform piezoelectric scaffolds. Cyclic compression of a hydroxyapatite/BaTiO_3_ piezoelectric scaffold, for instance, promoted the growth of osteoblasts [214] and stem cell chondrogenic differentiation compared to simple mechanical loading [215]. Yet this requires scaffold patterning to achieve localized stimulation. Finally, magnetic materials are also used to generate dynamic, tunable, and targeted mechanical stimuli in the engineered tissues [216]. Fluctuating magnetic fields near the embedded particles have been shown to enhance cellular processes and lead to faster tissue healing [217,218]. However, much like thermoplasmonic NPs, this effect is also believed to arise due to mechanical stimulation of the tissue in combination with a mild heating effect. Temperature increments up to 7 °C were measured for a PCL scaffold containing 10% (*w*/*w*) superparamagnetic iron-doped hydroxyapatite NPs after 5 min in a 30 mT alternating magnetic field [219].

Hybrid materials comprising iron oxide and mesogens, combined with an elastomeric matrix, have been used to generate reversibly deformable elastomers for dynamic cell culture applications. In the composite scaffold, deformation results from a nematic to isotropic phase transition triggered by the heating of the scaffold through the application of an alternating magnetic field. The inclusion of the nanomaterial also provides avenues to tune the properties of the composite, such as a variation in the phase transition temperature and an increase in Young’s modulus of the material [220]. In another work, dynamic alignment of magnetic NPs (magnetite spheres, iron oxide nanorods, and iron oxide spheres) induces up to 10-fold scaffold stiffening, generating forces to the encapsulated MCF10A cells comparable to those at the cellular level [221].

In fact, magnetic tissue stimulation was applied in numerous regenerative medicine applications for accelerated healing, targeting most notably bone [217,218], dental pulp [222], or muscle [223,224]. In the field of bone repair, magnetic stimulation combined with superparamagnetic NPs has been demonstrated to accelerate osteoblasts differentiation, enhance cellular adhesion and proliferation, and osteogenic differentiation in mesenchymal stem cells [61,225]. This approach can also be used in vivo when the patient’s body is unable to provide the necessary stimulation for healing to occur, for example for bone, cartilage, or nerve regeneration [21,226,227,228]. Moreover, magneto-mechanical transduction has been shown to activate mechanosensitive ion channels [229,230,231]. The application of magnetically triggered external forces on a large population of neurons using a magnetic hyaluronic acid matrix led to the activation of mechanosensitive ion channels PIEZO2 and TRPV4, as quantified from the calcium influx [232]. In many studies, alternating magnetic fields have been applied to wide regions of living tissues in combination or not with implanted nanomaterials. In contrast, a fluctuating, localized, and/or patterned magnetic field or excitation would enable targeted and selective excitation, promotion of growth and healing along a defined axis, the application of gradients of stimulation, etc. However, whereas optical triggers can be easily patterned, facilitating the delivery of localized mechanical stimulation to single cells [209], magnetic forces, in comparison, are more difficult to pattern, leading to non-localized, bulk stresses [61] or requiring heterogeneous, patterned NP scaffolds, to tune the location of the mechanical stimulation. The former approaches have been extensively used recently in the construction of magnetic bio-actuator devices [62,233,234,235].

Thus, in principle, depending on the combination of nanomaterials and physicochemical fields, specific cell receptors could be targeted by specific stimulation strategies. The positioning of nanoactuators respective to the cell membrane is a critical parameter that should be precisely tuned, and cellular internalization of the nanomaterial controlled. Depending on this positioning, electrical cell activity can, for example, be inhibited or enhanced. Another challenge is that, in general, multiple stimulation modalities can emerge in the excited particles’ surroundings. For example, light stimulation of plasmonic particles will lead to local heat generation in the particle vicinity, but also to the production of reactive oxidative species and high electromagnetic field enhancements. The heating may influence cell receptors, but also denature or deform ECM components, leading to changes in the mechanical properties of the matrix or even considerable shrinking or expansion of the surrounding hydrogel material, giving rise to significant mechanical stimulation. Another challenge that is valid for any hybrid material, is the distribution of the nanomaterial in the polymer or hydrogel matrix. Local heterogeneities in nanomaterial concentration lead to variations in the desired effect, for example, when light is used for nanoparticle stimulation, high nanomaterial concentrations will decrease drastically the amount of radiation available for excitation in the core of the construct, while insufficient nanomaterial will lead to an insufficient effect. For this reason, the effect targeted should be calibrated with respect to the stimulus intensity, and continuously monitored by means of local probes, which can also be chosen from the wide nanomaterial toolbox currently available, as will be discussed in the following section.

## 4. Nanosensing Functionalities in Engineered Extracellular Matrices

Developing functional artificial tissues requires also in situ tissue monitoring solutions, i.e., sensor modalities reporting cell functions, cell-matrix interactions, and/or environmental parameters, ideally, both at the bulk scale and especially at a single-cell level as in 3D models, concentration gradients are locally spatially varying [236]. In situ continuous monitoring is not present in most systems. Typically, cell morphology, viability, and proliferation are assessed by using optical or confocal fluorescence microscopy. Yet, although important, observation of morphological parameters provides (limited) information about cellular behavior, and fluorescence imaging often involves irreversible cellular damage, providing only end-point information. Long-term, non-invasive monitoring of tissue functions is highly desirable, through observation of nutrients, pH, temperature, electrical activity, secreted molecules (e.g., ammonia, etc.) metabolites (e.g., glucose), oxygen levels or lactate, reactive oxygen species (ROS), etc.

Many nanoparticle biosensors have been demonstrated, targeting mainly point-of-care diagnostic testing in body fluids, such as urine, saliva, or blood [237,238,239,240]. Recently, though, biosensors have also become an emerging component of TE for the detection of specific biological signature molecules. These sensors often rely on bio-/electrochemical or optical transduction schemes. Micro-and nanoparticles are particularly well suited for optical biosensing. They can fulfill either the role of transducer or in some cases both recognition element and transducer at once, alleviating the need for biological receptors such as antibodies or enzymes. Moreover, unlike electrochemical transduction, optical methods (fluorescence (quenching), localized surface plasmon resonance (LSPR), evanescent waves, etc.) allow for remote, localized monitoring of the (bio)chemical cell micro-environment, such as pH, the concentration of oxygen or ROS, secreted molecules, or physical quantities such as local temperature (nanothermometry) or mechanical stress. Compared to electrochemical sensors, NPs allow for higher spatial resolution in monitoring cellular and matrix parameters and, particularly, are more suitable for adaptation to 3D systems [241].

In the following paragraphs, we discuss the recent advances in using micro- and nanoparticles for monitoring TE constructs.

### 4.1. Monitoring the Chemical Micro-Environment

#### 4.1.1. Metabolic Activity Sensing

The chemical signature of the cell microenvironment provides paramount information on cell physiology. A metabolically active cell takes up energy (glucose, fatty acids) and oxygen and produces ROS and waste products. Hence the most basic, yet relevant parameters to measure cell metabolic status are pH, O_2_, ROS, and glucose. The integration of advanced in vitro models in drug efficacy tests or tumor progression in cancer research demands such accurate metabolic analysis. Especially gradients and local discrepancies in 3D are of interest, as it was shown that O_2_- and glucose-deprived stem cells displayed severely compromised functional maturation [242]. Particularly, non-vascularized 3D tissues often rely on molecular diffusion as a route for oxygen to reach the center of the tissue. As it is actively consumed before it reaches the inner cells, local hypoxia often results in cell death of the inner core tissue. In other scenarios, lower-oxygen levels might be desirable, as in vivo physiological oxygen levels can vary from up to 14% in lung alveoli down to 3% in muscle or even as low as ~0.5% when studying some cancers, tissues necrosis, or cardiovascular diseases [243]. Moreover, ROS molecules, as byproducts of oxygen metabolism, have an important role in cell signaling, neurotransmission, and many more [244,245]. Yet, the overproduction of ROS due to oxidative stress can significantly damage cellular structures. Nevertheless, so far little effort has been dedicated to the mapping of spatial and temporal gradients in 3D culture systems to establish precise control over local nutrient and ROS concentrations.

Traditionally, pH, O_2_, and ROS sensing in 2D cell culture is carried out using electrochemical methods (e.g., ion-sensitive electrodes, field-effect transistors, electrochemical impedance spectroscopy (EIS), etc.) [246]. Also, enzyme-based sensors rely on electrochemical transduction, for example using glucose-oxidase to convert glucose into other moieties while creating H_2_O_2_ [247]. Such electrochemical sensors rely on physical contact with electrodes. For instance, in the case of potentiometric sensors [248,249], which can be used for dynamic mapping of 2D pH variations [250], contact with the sensing electrode is required. Integrating such sensors into 3D cultures requires either the invasive insertion of (an) electrode (e.g., microneedle-shaped) probe(s) [251,252] or placing the 3D cell culture on top of a planar electrode surface. In the latter category, Shaibani et al., created an electrospun poly(vinyl alcohol)/poly(acrylic acid) (PVA/PAA) nanofiber hydrogel on top of a light-addressable potentiometric sensor surface and demonstrated the detection of changes in the pH of the medium used for cancer cells MDA MB231, MCF10A, and MDA-MB-435MDR cultures [253]. Alternatively, the growth of human hepatoma cells (Hep2G) in Matrigel and colony formation of Huh-7 cancer cells were monitored using EIS in a measurement cell with parallel, opposing planar electrodes [254,255]. Unfortunately, methods like EIS are also affected by environmental changes in the tissue construct such as hydrogel degradation or cell migration. A strategy to increase the compatibility of hydrogel systems with electrochemical sensing methods is to render the fibrous scaffold conductive by incorporating electrocatalytic capabilities [256]. Wu et al. and Zhang et al. fully modified a 3D, polyaniline, and poly(3,4-ethylenedioxythiophene (PEDOT)-coated polydimethylsiloxane (PDMS) scaffold, respectively, with platinum NPs to perform real-time detection of H_2_O_2_ [256,257]. The authors report detection limits of 1.6 µM in 0.01 M phosphate-buffered saline (PBS) [256] and 76 nM [257] with a linear response from 10 µM to 10 mM and 0.2 µM to 20 µM (sensitivity of 460 nA/µM), respectively. These scaffolds are shown in Figure 6. Similarly, GNP-decorated graphene and chitosan scaffolds have been designed for the detection of nitric oxide (LOD 9 nM, linear range 0.2 µM to 6 µM) released by mouse skin cells (JB6-C30) and tumor cells (B16-F10) [258] and H_2_O_2_ (detection limit 15 nM, linear range 0.1 µM to 1 mM with a sensitivity of 1.2 mAmM^−1^cm^−2^) [259], respectively. Yet, despite the high sensitivity such electrochemical devices only provide global chemical read-out. Electrochemical methods are difficult to adapt toward 3D spatial mapping of the chemical signature.

For 3D spatial mapping of chemical gradients, optical methods based on fluorescence [243,260,261] or phosphorescence [262,263,264] are preferred. In combination with current improvements in microscopy (photoactivated localization microscopy (PALM), stimulated emission depletion microscopy (STED), etc.), it allows reconstructing 3D tissues with high spatial resolution. In this sense, micro- and NP-based luminescence sensing has gone through astonishingly fast growth [265]. A popular strategy is to encapsulate O_2_-, pH-, or ROS-sensitive fluorescent molecules into micro- or NPs to create micro/nanoparticles for direct or ratiometric optical sensing [243,266,267,268]. For instance, Moldero et. al. loaded the pH-sensitive dye seminaphtharhodafluor (SNARF-I) in calcium carbonate (CaCO_3_) microparticles (3.5–5 µm) [266]. The emission of the SNARF-I loaded particles strongly depends on the pH, emitting green light in acidic and red light in basic conditions. These particles allowed for 3D ratiometric mapping (I_595 nm_/I_640 nm_) of the pH of human mesenchymal stem cell 3D cultures grown in poly(ethylene oxide terephthalate)/poly(butylene terephthalate) (PEOT/PBT) scaffolds for up to 7 days. These measurements revealed the existence of pH variations in the 3D scaffold that are more prominent compared to the 2D counterpart. Specifically, more acidic values were found within the scaffold with respect to the 2D culture. In another work, PDMS microparticles contained a silica gel holding the O_2_-sensitive (Ru_2_(Ph_2_phen_3_)Cl_2_) and oxygen-insensitive dye Nile Blue. The particles were capable of reversibly reporting the O_2_ pressure through their fluorescence intensity [269]. The fluorescent components can be replaced eventually completely with NPs to improve the long-term resistance to the photobleaching of organic molecules. This was illustrated by Poly-styrene-co-maleic anhydride (PSMA) encapsulation of the O_2-_sensitive luminescent indicator platinum(II) meso(2,3,4,5,6-pentafluoro)phenyl porphyrin (emitting red light) in combination with the inert fluorescent dye Bu3Coum (emitting green light) [260] or by micelle encapsulation of supramolecular assemblies of quantum dots with palladium(II)porphyrins [261]. Both were shown useful for simple, rapid, and non-invasive mapping of the chemical microenvironment and activity of encapsulated cells. Moreover, in the latter the large two-photon absorption cross-section of the QDs [261,270] was exploited to only excite QD-bound porphyrin through Förster resonance energy transfer (FRET), eliminating any background signal from free porphyrins, while the QD emission is unaffected through O_2_ changes, serving as the ratiometric standard for the O_2_-sensitive porphyrin emission. FRET was also employed by Park et al., who encapsulated molybdenum disulfide (MoS_2_) nanosheets in microcapsules with an aqueous core and a semipermeable polymeric shell. The MoS_2_ nanosheets were functionalized with pH-responsive polymers having fluorescent groups at the distal end to provide pH-sensitive FRET [271]. Instead of encapsulation within a micro or nanoparticle, Chu et al. functionalized the surface of silicon NPs with pH-sensitive dopamine and non-sensitive Rhodamine B isothiocyanate to perform ratiometric pH sensing for intracellular pH sensing [272].

Apart from measuring and manipulating O_2_ profiles, several molecular compounds have been suggested for the optical detection of ROS as well [273,274,275]. One method involves using hydrocyanines, whose fluorescence substantially increases upon oxidation by superoxides or hydroxyl radicals [274]. For the detection of H_2_O_2_, Kim et al., designed ROS-sensitive nanoprobes called ‘nanopebbles’ by incorporating the fluorescent molecule 2′,7′dichlorofluorescein within organically modified silica nanoparticles, thereby blocking the interference of other ROS species and enzymes (peroxidases for instance) [276], the NPs were finally delivered into the cytosol of RAW264.7 macrophages through conjugation of the membrane penetrating cysteine terminated TAT peptide to the NP surface. The NPs proved successful in measuring H_2_O_2_ concentrations in the 10 nM to 100 µM range. Selectivity for hydrogen peroxide was also achieved by employing peroxalate nanoparticles containing a fluorescent dye. The reaction of H_2_O_2_ with the peroxalate esters results in the chemiluminescence of the trapped dye molecules, excited by the generated dioxethane intermediate [277]. Table 1 displays a comprehensive list of recently developed fluorescence-based micro-/nanoprobes for the optical detection of pH, O_2_, and ROS levels for 3D TE. Several probes have been explored for their ability to map the 3D chemical cell environment, both spatially and temporally [260,261,262,263,266,278]. Yet, implementation remains challenging, and retrieved data is often still restricted to in-plane information with limited depth-profiling or 3D reconstruction.

Despite the developments described in this section, the implementation of optical methods in combination with NPs for monitoring 3D tissue structures still provides several challenges, especially for long-term experimentation. The most important limitation involves the required high light intensities, which induce photobleaching and cell phototoxicity [279,280]. Phototoxicity can lead to obvious cell damage or more subtle changes in cell functions, especially occurring during extended periods of continuous imaging. In particular, 3D mapping often requires high spatial and/or temporal resolution (e.g., electrical activity recordings), both requiring high light intensities in order to generate a sufficient signal. When possible, light intensities should be reduced for longer exposure times, or alternatively, two-photon microscopy can be used as longer excitation wavelengths scatter less in biological tissue. Also, pulsed or intermittent illumination has been shown to effectively reduce phototoxicity and reduce probe photobleaching. Secondly, most reported fluorescent NP probes did not display adverse effects on cell viability, yet potential NP uptake, long-term toxicity, and more subtle effects on cell functioning and morphology must still be investigated by performing adequate control experiments, as it can potentially lead to incorrect interpretation of experimental results. As seen from Table 1, few long-term studies have been performed with such nanosensors.

**Table 1 gels-09-00153-t001:** Recently developed fluorescent nano-/microprobes for sensing O_2_, ROS, and pH that are applied intracellularly (*), extracellularly in 2D monolayer cell cultures (♦), 3D mapping (†), in vivo (‖) and long-term (>7 DIV) (§).

Target	Fluorescent Nano/Micro-Probe	Ref.
pH ^†§^	Carbon QDs	[281]
pH *	Silicon NPs coated with dopamine and Rhodamine B isothiocyanate	[272]
pH ^†^	SNARF-1 encapsulated in CaCO_3_ NPs	[266]
pH ^♦^	silicon microcapsules containing molybdenum disulfide (MoS_2_) nanosheets coated with rhodamine B terminated DEA and BMA	[271]
pH ^†^	Polyurethane NPs containing fluorescein and diphenylanthracene	[267]
pH	NaYF_4_:Yb,Er upconversion NP	[282]
O_2_ *	Pd-tetra- (4-carboxyphenyl) tetrabenzoporphyrin dendrimer and Alexa 647 in polyacrylamide NPs	[268]
O_2_ *^‖^	Mesoporous silica containing NPRu(dpp)_3_]^2+^Cl_2_ with an upconversion nanoparticle	[283]
O_2_ ^†^	Ru(dpp)_3_Cl_2_ (C_72_H_48_Cl_2_N_6_Ru) immobilized on a polystyrene NP	[263]
O_2_ *^§^	phosphorescent Pt(II)-tetrakis(pentafluorophenyl)porphyrin and poly(9,9-diheptylfluorene-alt-9,9-di-p-tolyl-9H-fluorene) encapsulated within poly(methyl methacrylate-co-methacrylic acid)-based nanoparticles	[264]
O_2_ ^†^	palladium-benzoporphyrin in porous poly(2-hydroxyethyl methacrylate) gell particles	[262]
O_2_ ^†^	tris (4,7-diphenyl-1,10-phenanthroline) ruthenium (II) dichloride, or Ru(Ph_2_phen_3_)Cl_2_ with Nile blue chloride in PDMS	[269,278,284]
O_2_ ^†^	PSMA encapslation of platinum(II) meso(2,3,4,5,6-pentafluoro)phenyl porphyrin and Bu_3_Coum	[260]
O_2_ ^†‖^	Micelle encapsulated palladium(II)porphyrins with QDs	[261]
H_2_O_2_ *	DCFDA in an Ormosil NP	[276]
H_2_O_2_ ^‖^	peroxalate NP containing pentacene	[277]
HO *	carboxyfluorescein-DNA-GNP	[285]

Microscopy techniques reducing the excited volume by limiting the illumination of the sample to the focal plane and or algorithms dealing with image reconstruction under low illumination conditions are constantly being developed and improved [279,286]. In 2D, relevant for functional surfaces, TIRF (total internal reflection fluorescence) for instance, relies on the evanescent field only penetrating the sample for a distance of a few hundred nanometers to excite a small volume. In 3D, light-sheet fluorescence microscopy or two-photon fluorescence microscopy, with its inherent optical sectioning and high penetration depth allow for 3D reconstruction with high spatial resolution [280].

As a final remark, long-term biological tissue imaging also imposes stringent environmental and hardware conditions. Samples must be constructed in a fast and sterile fashion and kept in an environmental enclosure at optimal temperature, humidity, and O_2_ concentration. For time-lapse experiments, the medium must be optimized for low auto-fluorescence and evaporation and perfusion should be controlled as well as the effects of humidity and temperature-induced drifts on the imaging optics.

#### 4.1.2. Other Signaling Biomarkers

Apart from metabolic activity, it is desirable to precisely monitor the secretion of specific, small messenger molecules to validate realistic in vitro biomaterial-based tissue models. Biomarkers are indicative of tissue physiology, and monitoring their levels provides information on the development of a well-functioning tissue. Many diagnostic biosensing devices have been proposed for the detection of biomarkers in complex body fluids like blood, saliva, or urine, such as troponin and creatine kinase as indicators for myocardial infarction [287], C-reactive protein (CRP), and P-selectin for ischemic stroke, CA 19-9 and B-type natriuretic peptide (BNP) for pancreatic cancer or pro-inflammatory cytokines and chemokines, to list a few.

Protein biomarkers are typically detected through antibody-antigen (Ab-Ag) coupling, often in combination with flow cytometry, enzyme-linked immunosorbent assays, or immunofluorescence. The transduction systems, most compatible with real-time detection of secreted biomolecules in 3D matrices would be based on (L)SPR, SERS (surface-enhanced Raman scattering), or even mass spectroscopy. Approaches to monitoring the global presence of secreted markers in cell cultures involve 2D planar sensing surfaces that support 3D tissue structures on top [288]. Berthuy et al., cultured human prostate carcinoma cells in 3D micro alginate gels positioned on top of a gold substrate functionalized with Ab sensing spots for *β*_2_-microglobulin (*β*_2_M) and prostate-specific Ag [289]. The binding of these biomarkers, indicative of phases of prostate cancer, was subsequently detected using SPR. The technique provides information on the release by an ensemble of cells in a gel, but by being diffusion limited, is not suitable for real-time, continuous, long-term kinetic monitoring, due to fast saturation of the sensing surface. In another work, repeated sensor regeneration was achieved for a label-free electrochemical detection surface and applied to albumin, GST-*α,* and creatine kinase-MB release by primary hepatocytes in GelMA [290] or IL-6 and TNF-*α* secreted by 3D muscle microtissue [291]. In a more 3D-based approach, the planar gold surface can be exchanged for Ab functionalized GNPs using the LSPR as a detection scheme. Or other particles such as fluorescently labeled beads [292] or CNTs [293] can be combined with Abs, aptamers, or DNA strands as recognition elements. One example involves an aptamer-coupled GNP-QD assembly, where the interaction of the aptamer with cardiac troponin leads to a fluorescence enhancement through FRET [294]. These nanobiosensors, unfortunately, have been limited to controlled microfluidic devices, and are yet to be incorporated into complex tissues. Integration in complex microgel systems has mainly been hampered by sensor regeneration, as in microfluidic chips fresh NPs can be introduced after washing steps [295] as label-free nanobiosensors, that serve both for biomarker detection and signal transduction through LSPR or SERS. SERS takes advantage of the excitation of LSPRs during Raman scattering from molecules adsorbed on nanostructured metal surfaces because each probe molecule presents a unique vibrational spectral fingerprint. Although SERS spectra in complex systems are often difficult to interpret and translate into the correct chemical or biological footprint because of the presence of many heavy molecules that easily adsorb on the metallic NPs, SERS spectroscopy is well suited for non-invasive monitoring. Plou et al. incorporated GNRs within a 3D bioprinted gelatin-alginate hydrogel scaffold for SERS-based drug diffusion monitoring [296]. The technique allowed tracking the diffusion of the drug methylene blue [296] as well as the biomarker adenosine [297] throughout the 3D printed scaffold. To illustrate the versatility of the technique for in situ SERS-based detection of cell differentiation, Kim et al. utilized the fact that the membrane of undifferentiated neural stem cells is generally comprised of poly-unsaturated molecules, rich in C=C bonds characteristic of aromatic structures [298]. Graphene-oxide-encapsulated GNPs give an enhanced SERS signal only for undifferentiated stem cells as the graphene acts as a SERS enhancer by interacting with aromatic structures through *π* − *π* stacking. Similarly, El-Said used SERS to in situ distinguish the cell-specific differentiation footprint of PC12, embryonic, and adult stem cells cultured in real-time on a gold nanostar-coated surface [299].

Nevertheless, the use of GNP LSPR sensors in TE still provides a lot of challenges. Most notably, the regeneration for continuous and long-term monitoring in complex tissue constructs is a major problem. High selectivity and sensitivity for the target analyte in combination with good sensor reversibility are essential. Also, the uncontrolled desorption and/or degradation of biological recognition elements (enzymes, aptamers, DNA, Abs, etc.), as well as the NP uptake by cells represent major challenges with respect to long-term and reversible chemical monitoring.

### 4.2. Sensing Endogenous Physical Fields

While the previous section focused on strategies for mapping the chemical microenvironment of 3D tissues, similar efforts are being made to develop sensors revealing endogenous physical fields in 3D cell cultures, such as mechanical stress, temperature, and electric fields. Most of the developments in this area are still focusing on probe development and calibration rather than on the final application, i.e., mapping fields created by cellular activity. In the following, we will discuss methods for the monitoring of physical properties in the extracellular domain. Intracellular monitoring based on nanoparticle uptake falls outside the scope of nanocomposite cell scaffolds.

#### 4.2.1. Temperature

NP-mediated heating plays a crucial role in thermotherapy, for instance, by inducing tumor cell death by hyperthermia, dynamic drug delivery, or even by promoting wound healing [161]. Hence, the mapping of temperature in tissues is a key aspect of nanomedicine. Typical tools for in vivo temperature imaging are MRI thermometry [300,301]—potentially using temperature-sensitive magnetization of a nanomaterial contrast agent [302]-, and photo-acoustic mapping [303]. Moreover, diagnostic methods heavily rely on thermal imaging as the onset of many diseases, e.g., inflammation, cancer, and cardiac abnormalities, is characterized by thermal singularities. For in vitro thermal mapping at single-cell resolution, near-infrared cameras [207], scanning thermal microscopy [304] or optical diffraction tomography [305], and luminescent temperature-sensitive nanoprobes are being researched. Optical imaging using thermosensitive fluorescent NPs offers an excellent alternative for fast, real-time monitoring using (multimodal) imaging techniques [306]. They enable remote, optical detection through various temperature-sensitive luminescence properties such as intensity, lifetime, and spectral shifts with high spatial and thermal resolution [307,308]. As such nanothermometry research has resulted in the development of a plethora of temperature-sensitive fluorescent probes including organic dyes or (dye-based NPs [211,309,310,311,312], metal-organic frameworks, or polymers with rare-earth centers [313,314,315,316], up- and downconversion particles [317,318], nanodiamonds with nitrogen vacancies [319,320], as well as metal and semiconductor NPs [162,321,322,323,324]. The working principle for these dyes is described in detail in the recent review from Bradac et al. [325].

In the case of organic dye-based temperature sensing, the thermal quenching of the excited fluorophores is often used as a measurement mechanism. The variation in the decay rates from the fluorophore excited state as a function of temperature leads to measurable changes in both fluorescence intensity and lifetime. Other dye-based mechanisms explored for fluorescence thermometry include monomers–excimer, monomers-exciplex, or twisted intramolecular charge transfer fluorescence [325]. The magnitude of the thermal sensitivity varies from dye to dye [312]. Rhodamine, belonging to the group of xanthenes, is among the most popular dyes, as a result of its superior chemical stability as well as its relatively high luminescence efficiency. Silica-encapsulated rhodamine NPs have been incorporated into the 3D plasmonic hydrogels to monitor local temperature changes induced by plasmonic heating (see Figure 5A,B) [211]. These NPs exhibited a stable and reversible temperature sensitivity of approximately 1%/°C within a temperature range of 20 to 50 °C. Alternatively, metal-organic-frameworks (MOFs) with lanthanide centers are also fluorescent and display a temperature-dependent fluorescence intensity. Such lanthanide-based nanothermometers have been applied to measure thermogenesis in Hela cells detecting temperature variations of 1 °C in the physiological range [310]. In this work, the intensity reduction rate was found to be close to 3% and hence superior to fluorescent dyes. Rare-earth ions are also the basis for temperature-sensitive upconversion nanoparticles (UCNP), which offer high fluorescence stability, high quantum yield (QY), and narrow emission bands that cover the entire electromagnetic spectrum. MOFs are excellently suited for integrated ratiometric sensing through the inclusion of different light-absorbing groups and emission centers. The combination of Eu^3+^ (612 nm emission) with Tb^3+^ (545 nm emission), for instance, allowed for ratiometric sensing with temperature sensitivities close to 4.9%/°C (about 5 times higher than sensitivities of typical temperature sensitive fluorophores) [326]. In another work, Ma et al., synthesized Nd-doped silicate glasses and incorporated the temperature-sensitive bioglasses in injectable alginate hydrogels for localized temperature sensing at tumor sites [327].

Finally, the emission band of semiconductor quantum dots is strongly modulated by temperature. Especially the strong decrease in fluorescence intensity upon temperature increase is remarkable [328]. This is attributed to the activation of phonon-assisted processes and thermally assisted energy transfer from the bulk to the non-radiative surface states. Moreover, it was shown that the intensity-based temperature sensitivity of CdSe QDs is enhanced in two-photon fluorescence excitation with respect to single-photon excitation, while the wavelength shift remained unaffected [329]. The most sensitive fluorescence nanothermometer to date was recently reported by Wang et al., who synthesized yellow emitting carbon QDs with a temperature sensitivity of up to 5.3%/°C and a resolution of 0.09 °C [330].

Most of the developed nanothermometers were employed for either intracellular imaging or in vivo injection into animal tumor sites. Literature reports on the incorporation into artificial ECMs, for localized temperature monitoring of developing tissue with the purpose of researching cellular responses/properties as a function of temperature, are rather limited. In vitro, thermal sensing is nevertheless vital for understanding cell behavior. Temperature is a critical parameter influencing cell fate, growth, differentiation, or migration. Moreover, temperature influences the structural properties of most proteins, which can already denature when heated slightly above physiological levels. Therefore, effective and biocompatible nanothermometers would be very useful for the development of successful TE platforms.

#### 4.2.2. Mechanosensing

3D mechanosensing studies the mechanical interaction of cells with their environment, usually a hydrogel scaffold [331,332]. Measuring cell applied forces, in the nN range, is essential for understanding mechanotransduction and cell-ECM interactions [333,334,335]. Typical and well-established techniques for probing cell-exerted forces are traction force microscopy (TFM) and particle tracking microrheology [336]. In the former technique, micro-sized fluorescent particles are incorporated into the ECM material. As the cells exert forces, they displace the particles throughout the hydrogel while the position is tracked. From these images, the displacement field can be computed and translated into strain and stress fields given that the mechanical properties of the hydrogel are well characterized [337,338]. Huang et al., for instance, imaged the 3D distribution of stress exerted by growing B16-F10 cells in methacrylated alginate gels by embedding 200 nm fluorescent beads. They could follow dynamic growth-induced stresses in gels with varying degrees of stiffness through dynamic particle tracking [339]. Alternatively, although so far exclusively in 2D, techniques involving substrates with micropillars or other deformable structures have been developed (e.g., suspended monolayers [340] or microbulges [341]), as shown in Figure 5E,F where the bending of the micropillars is used to calculate the stress exerted by the cells on top [207,336]. Microrheology, on the other hand, investigates the viscoelastic properties of (components of) the cell cytosol by tracking the displacement of intracellularly injected tracer beads. A thorough discussion of the above methods is beyond the scope of this review, i.e., nanocomposite ECMs. The reader is referred to other relevant review articles [335,342].

To accurately measure nN and even smaller forces and induced displacements, optical nanoprobes are continuously being developed. (Intra)cellular tension sensors are mainly based on FRET, allowing us to study interactions between structural proteins, cell-cell interactions, as well as cell-substrate interactions [336], while extracellular probes take shape as NPs that can be added as filler elements to the artificial ECM. Raja et al., for instance, developed CdSe/CdS tetrapod quantum dot stress sensors. These probes were incorporated in a poly-L-lactic acid electrospun polymer matrix [343,344,345] up to 20% wt. The fluorescence of the encapsulated tetrapods could report the elastic and plastic regions of polymer deformation during extension and compression without significantly changing the mechanical properties of the NP-loaded fibers and remained stable for many mechanical test cycles before failing. Such probes could potentially be incorporated into 3D matrices for locally sensing cellular adhesion and local stresses exerted by cells. Another strategy employs mechanosensitive upconversion nanoparticles, where the strain of the lattice results in a change in fluorescence lifetime, line width, and intensity [346,347,348,349]. Through control of the lanthanide dopant, the emission colors can be tuned, and ratiometric force sensing of somewhat larger, µN forces could be demonstrated [347,350]. Despite promising advances and high sensitivity, the practical application of fluorescence-based stress and strain sensors is mainly limited to testing polymer strength for civil engineering applications and is not implemented in TE [351,352].

#### 4.2.3. Electric Activity

Electrophysiological measurements are of huge importance for heart and brain TE, and a vast number of resources are invested in miniaturization (to micro or nano-scale) [353] and integration of electrophysiological techniques with 3D extracellular matrices [354,355].

NPs with electric-field sensitive optical properties can potentially present themselves as rivals for flexible mesh electronics [356,357], overcoming potential issues in connectivity that may arise in 3D geometries. In Section 3.2 we discussed how semiconductor NPs, plasmonic GNPs, and magnetic NPs could generate electric fields (or temperature) to remotely stimulate electrogenic cells through the opto-electric, opto-thermal, and magneto-electric effects, respectively. The reverse, where the transmembrane electric field impacts the optical or magnetic properties of the NP is also possible. The LSPR of GNPs, for instance, are sensitive to changes in the refractive index of the surrounding medium. The principle has been illustrated for neurons cultured on an array of 160 nm sized GNPs with the NPs reporting stimulated APs through the addition of glutamate to the culture medium [358]. The direct effect of electric fields on QDs optical properties (intensity modulation, spectral shift, and lifetime change), on the other hand, has been actively researched, with so far relatively low success rates in cellular preparations, likely due to the difficulty of realizing stable QD membrane insertion [359,360,361]. More successful approaches involve QD-based two-component systems, utilizing Förster resonance energy transfer (FRET) or electron transfer (ET), with a second probe located in the lipid bilayer [362,363]. Finally, magnetic NPs have been suggested as magneto-electric probes for wireless mapping of neuronal electric activity using magnetic particle imaging, which maps the magnetization response of the NPs with high spatial resolution [364,365,366]. Yet, unlike neuronal stimulation, this has not yet been demonstrated experimentally.

Optical sensing of cell electric activity is particularly suited for 3D systems, as, unlike electrical methods, no direct physical contact is required with the tissue. Moreover, light-based recordings offer excellent spatial resolution (subcellular) and simultaneous recordings at various locations. Well-established, successful reporter probes, that convert the membrane potential into an optical signal, are typically (1) voltage-sensitive dyes that are inserted within the leaflets of the cell membrane, (2) genetic voltage indicators, whose expression can be controlled and targeted to specific (sub)types of cells or even sub-cellular domains and (3) calcium indicators. The best probes offer sensitivities up to almost 50% per 100 mV, but they suffer from photobleaching, damaging the probe and limiting the duration of the experiment, and photo-damage, affecting the health and functioning of the targeted cells. Calcium recording, finally, is an indirect method, intracellular calcium release may occur without changes in membrane potential, and it is rather limited by the slow time course of intracellular calcium signals in response to AP generation [367]. Optimization of NPs as probes in combination with advanced microscopy techniques, like two-photon fluorescence imaging, can potentially overcome these limitations. QDs, for instance, have very large two-photon absorption cross-sections, 100 to 1000 times larger than voltage-sensitive dyes, which could allow optimizing fluorescence signal-to-noise ratios because typically only a limited amount of organic reporter probes can be embedded in the lipid bilayer and small numbers of photons detected. The combination with two-photon fluorescence also allows for fluorescence excitation using longer wavelength light, leading to less scattering and hence larger penetration depths, and optical section, as fluorescence is only generated at the focal point.

To summarize the key points of nanosensors in the context of cellular matrices, immobilization of nanoparticles with sensing modalities into cell culture scaffolds offers many advantages with respect to the use of free moving nanoparticles, which is already a major field of research. Once bound, the nanoparticles can be protected from endocytosis or removal from the measurement medium for a prolonged period, at least until their host matrix dissociates. When bound to a cell-adhesive matrix, the nanoparticles can be brought near the cells for sensing parameters such as markers of metabolic activity, voltage, or temperature, while preventing or delaying endocytosis. In combination with unbound particles targeted to enter specific cell types, these matrices offer a complete outlook on the phenomena occurring inside cell-laden scaffolds. Moreover, in contrast with histological techniques, nanoprobes can be used for real-time, quantitative cell culture parameter measurement. Different sensing or reference compounds can be included in a single particle, or multiple particle types can be used to enable ratiometric and multiplexed optical sensing schemes. With respect to electrode-based sensing techniques, optical sensing provides increased resolution and flexibility, in particular for 3D systems. They decrease the invasiveness associated with the use of electrodes, which can alter cellular environments, and reduce cell viability or inflammatory responses. As for matrix enhancement and stimulation roles, the accurate positioning of the nanoparticles is crucial when parameters can only be sensed in specific cell locations, such as cellular potential within nanometers of the cell membrane, cell-exerted forces at adhesion sites, or temperature fields that dissipate over micrometer distances from heat sources. Another challenge is the crosstalk that can occur between sensing modalities when measuring more than one parameter, for example, temperature and oxygen combined with voltage sensing. Furthermore, one nanosensor is often influenced by more than one parameter, for example, pH, O_2_, and temperature often influence multiple particle optical responses. Therefore, several types of particles must be used in this case, together with careful calibration and testing, coupled with specific acquisition and signal treatment strategies. Finally, for more complex sensing targets such as nucleotides, cytokines, or other proteins, typical limitations of biosensors similarly apply to nanosensors, such as fouling under physiological conditions, limited/slow reversibility of the molecular receptors, and batch-to-batch sensitivity variations, thus, solutions addressing these challenges are shared with the biosensors field.

## 5. Conclusions

Nanomaterials offer a highly versatile toolbox for enhancing the functionalities of tissue scaffolds. Their physical properties allow tuning the mechanical, optical, and/or electrical properties of a scaffold, and dynamically changing the scaffold structure in combination with external fields (shear flow, magnetic, or electric fields). Moreover, NPs can also report environmental changes through a change in their optical or magnetic properties. Their unique physical properties and high versatility make them ideal components for the construction of bidirectional cell matrices that are (1) recapitulating the physical properties of the in vivo environment and, at the same time, capable of (2) delivering dynamic cues to guide cell behavior and (3) read-out either the culture parameters (e.g., oxygen levels, pH values, temperature) or the local cellular responses (e.g., secreted biomarkers, electrical activity, etc.).

The remote addressability of functional NPs makes them excellently suited for 3D dynamic cell culture. In terms of stimulation, GNPs have been most widely employed as optical absorbers to convert light into a thermal stimulus. These thermal stimuli have been used to guide cell migration or evoke cell electrical activity in neurons and cardiac cells. Cell electric stimulation has also been done with QDs, where the exciton’s dipole moment can perturb the membrane potential.

Integrated NP sensors, on the other hand, have been the least explored in combination with hydrogel materials. A plethora of probes have been developed for the 3D mapping of oxygen levels and temperature. Yet, so far, only in a few studies, they have been integrated with hydrogel cell matrices and electrochemical sensors for the detection of ROS, LSPR- and SERS-based GNP sensors for secreted biomarkers, and flexible fishnet electronics so far have been used for revealing bulk information or low-resolution mapping. The development of NPs for the detection of biomarkers, for instance, is hampered by the difficult regeneration of the recognition element on the NP surface or the interpretation of complex spectra (e.g., SERS) in tissue constructs. The use of NPs for remote electro-optical or magneto-electric E-Field sensing has been suggested, but their implementation currently suffers from problems in achieving stable, long-term positioning with respect to the cell membrane and their potential for reliable, high-throughput electrical stimulation and sensing remains still to be realized.

## Figures and Tables

**Figure 1 gels-09-00153-f001:**
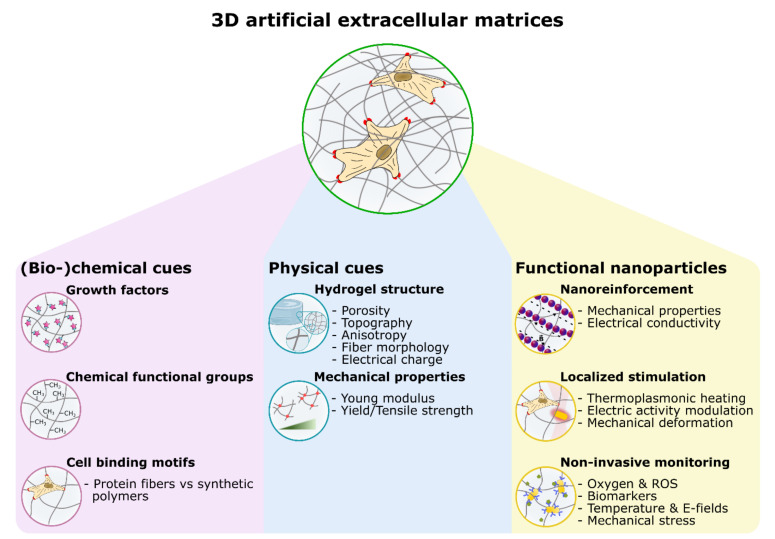
Material toolbox available for mimicking the cell microenvironment. Parts of this figure are adapted with permission from Ref. [6]. Copyright 2017 American Chemical Society.

**Figure 2 gels-09-00153-f002:**
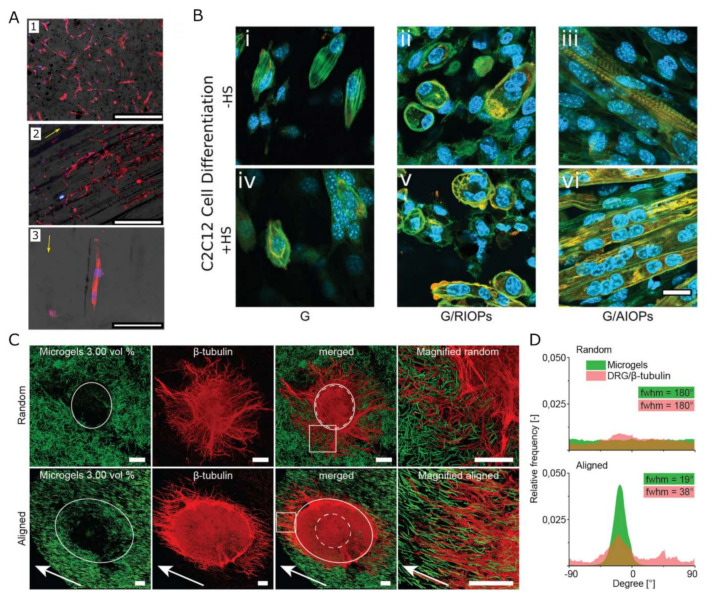
Directional cell growth on magnetically aligned nanocomposite hydrogels. (**A**) Fluorescence image of human mesenchymal stem cells on top of GelMA containing 1. non-oriented IONPs (scalebar 400 µm) 2. oriented IONP filaments (scalebar 400 µm) 3. oriented IONP filaments (scalebar 100 µm). (**B**) Fluorescence images of D API (blue) and phalloidin (green) stained C2C12 differentiated cells with (+HS) and without (−HS) horse serum, cultured on top of GelMA (G), GelMA with non-oriented IONPs (G/RIOPs), and GelMA with oriented IONP filaments (G/AIOPs) (scale bar 20 µm). (**C**) Fluorescence image of DRG (red, Alexa fluor 633) cultured in a fibrin hydrogel with 3 vol% of IONP-PLGA microgels (green, fluorescein) both randomly oriented and aligned (scale bar 200 µm). (**D**) Distribution of neurite orientation starting at the white full circle. The edge of the DRG body is marked by the white dotted circle. Panels (**A**,**B**) are reprinted with permission from Ref. [103]. Copyright 2019 Wiley. Panels (**C**,**D**) are reprinted with permission from Ref. [106]. Copyright 2017 American Chemical Society.

**Figure 3 gels-09-00153-f003:**
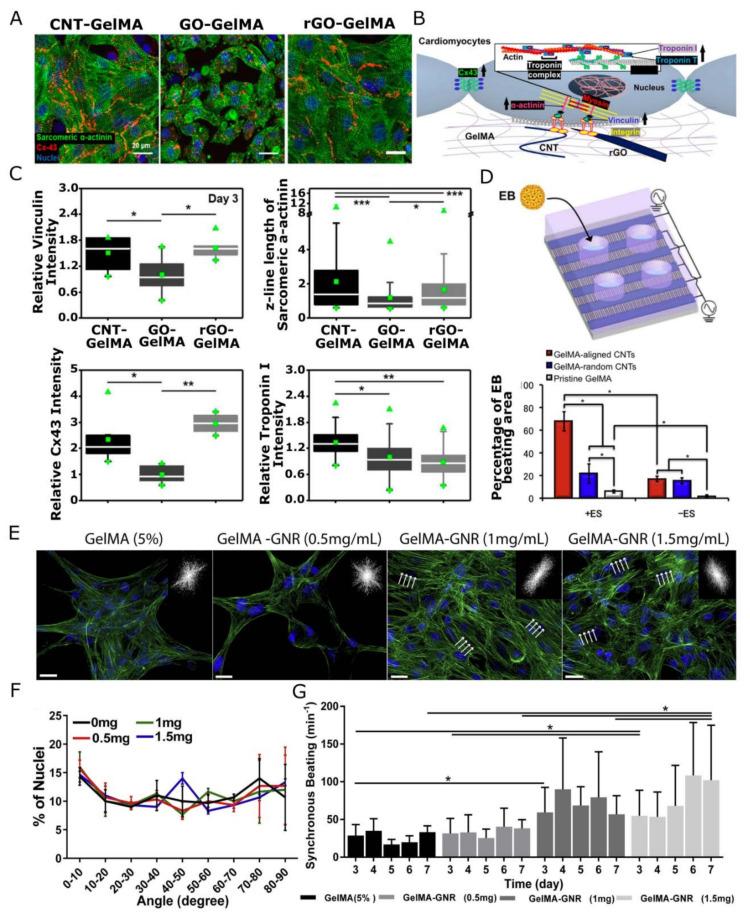
(**A**) Cardiac phenotypes on CNT-, GO-, and rGO-GelMA nanocomposite scaffolds. Fluorescence microscopy image of cardiomyocyte culture, labeled for sarcomeric α-actinin (green), Cx-43 (red), and nuclei (blue). (**B**) Schematic representation of important functional proteins during cardiomyocyte maturation. (**C**) Relative intensity of vinculin, z-line length of sarcomeric α-actinin, connexin-43, and Tropinin I after 5 days of culture as determined from microscopy images as shown in panel (**A**) (* *p* < 0.05, ** *p* < 0.005, *** *p* < 0.005). (**D**) Schematic illustration of stem cell culture of mouse embryoid bodies (EBs) in GelMA containing 0.5 mg/mL random or aligned CNTs with electrical stimulation (ES) of 1 Hz, 3V and the percentage of beating EBs on pristine GelMA and CNT-GelMA in the presence (+ES) and absence (−ES) of electrical stimulation after cardiac differentiation. (* *p* < 0.05) **E**) Fluorescence confocal microscopy image showing F-actin (green) and nucleus (blue) staining of cardiomyocytes cultured within GelMA and GelMA-GNR hydrogels for 7 days. The inset shows the Fourier transform, indicating local alignment of the F-actin fibers (indicated with white arrows). Scale bar 50 µm. (**F**) The alignment distribution for the nuclei from panel (**D**), indicating no global alignment of the nuclei within the (GNR-)GelMA scaffolds. (**G**) Synchronized beat rates from day 3 to day 5 for the various GNR-GelMA scaffolds shown in panel (**D**). (* *p* < 0.05) Panels (**A**–**C**) reprinted with permission from Ref. [133]. Copyright 2019 American Chemical Society. Panel (**D**) reprinted with permission from Ref. [82]. Copyright 2016 Elsevier. Panels (**E**–**G**) reprinted with permission from Ref. [137]. Copyright 2016 Elsevier.

**Figure 4 gels-09-00153-f004:**
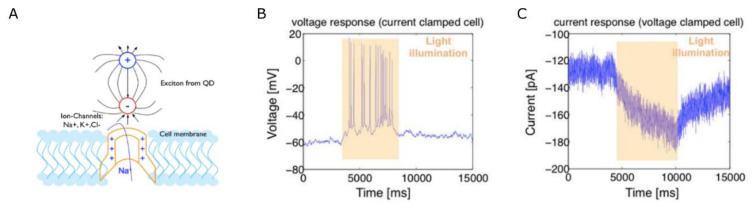
Opto-electric cell stimulation using QDs. (**A**) Schematic illustration of the interaction of a quantum dot with the cell membrane. (**B**) The voltage response of a current-clamped cortical neuron cultured on a CdSe QD film. (**C**) The current response of a voltage-clamped cortical neuron on a CdSe QD film. The figure is reprinted with permission from Ref. [187]. Copyright 2012 Optical Society of America.

**Figure 5 gels-09-00153-f005:**
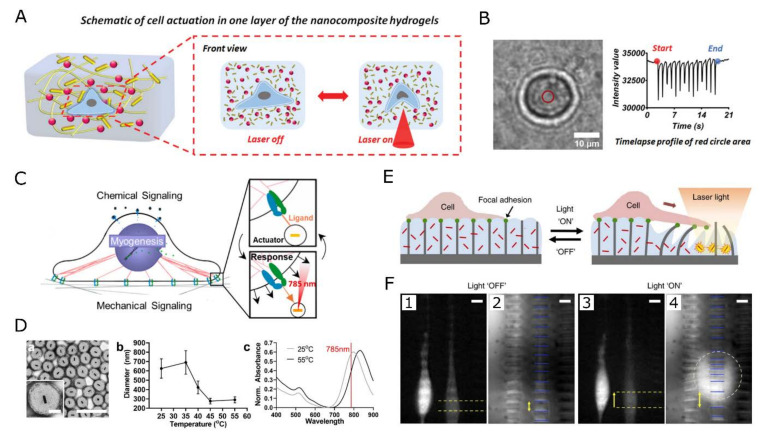
GNP-based optomechanical hydrogel actuation. (**A**) Illustration of a 3D GNR-PEG nanocomposite hydrogel for cell mechanical actuation and (**B**) an encapsulated SH-SY5Y cell ‘beating’ in response to 1 Hz NIR laser stimulation. (**C**) Illustration of the operating principle of PNIPMAM-encapsulated GNRs and (**D**) TEM-image of the NPs (Scale bar 1 μm, inset scale bar 200 nm), the hydrodynamic diameter as a function of temperature as measured by dynamic light scattering and the UV-vis-NIR absorbance spectrum at 25 °C and 55 °C with n red the NIR wavelength used for remote actuation. (**E**) GNR heating deforms a PNIPAM matrix exerting stress on the cell, cultured on top. The flexible nanowires embedded in the hydrogel translate the deformation into a force. (**F**) Epifluorescence images of the cells (1, 3) and brightfield images of the underlying microstructures (2, 4). The laser is focused at the center of the dashed circle that outlines the area where the hydrogel is contracted. Images (1, 2) indicate the start of the experiment, while (3, 4) are the same cells 3 s after 18 mW laser illumination. The locations of two microstructures at both timepoints are marked by the yellow dashed lines, and the elongation from 9 µm (2) to 13 µm (4) is indicated by the yellow arrows. Scale bars are 10 µm. Panels (**A**,**B**) are reprinted with permission from Ref. [211]. Copyright 2022 Wiley. Panels (**C**,**D**) are reprinted with permission from Ref. [210]. Copyright 2020 American Chemical Society. Panels (**E**,**F)** are reprinted with permission from Ref. [207].

**Figure 6 gels-09-00153-f006:**
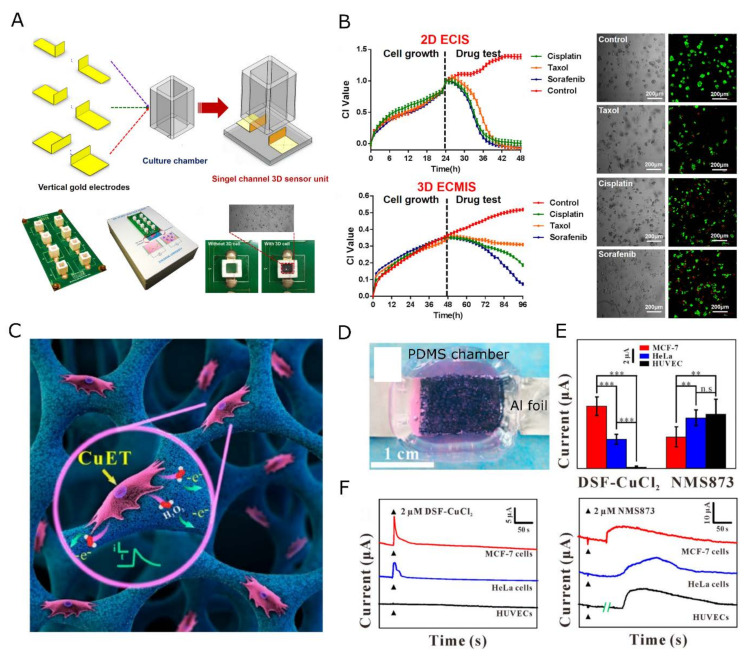
Electrochemical monitoring in 3D scaffolds. (**A**) Schematic representation of a 3D planar electrode impedance sensor for bulk monitoring of cell proliferation in 3D systems and (**B**) normalized cell growth curves as measured by 2D electric cell-substrate impedance sensing (ECIS) and 3D electric cell/matrigel-substrate impedance sensing (ECMIS) of 2D and 3D cultures of HpeG2 cells on top/within, treated with different 10 µL of the anti-cancer drugs Cisplatin, Taxol and Sorafenib. The cell index (CI) value represents normalized impedance change. The (fluorescence) microscopy images display cell morphology and live/dead staining of the different conditions in 3D ECMIS after 96 h. (**C**) Schematic representation of a PCP/Pt 3D electrochemical scaffold. (**D**) Picture of the PCP/Pt scaffold in cell culture. (**E**) Amplitude of the amperometric response of the PCP/Pt scaffold (vs. Ag/AgCl) upon the addition of 2 µL DSF-CuCl2 and NMS873 to MCF-7 (red trace), Hela (blue trace) and HUVEC (black trace) cells cultured for 5h on the electrochemical scaffold. (**F**) Transient amperometric response of the cells to 2 µL of the anti-cancer drugs DSF-CuCl2 and NMS873 corresponding to the data in panel (**E**). Panels (**A**,**B**) are reprinted with permission from Ref. [254]. Copyright 2019 Elsevier. Panels (**C**–**F**) are reprinted with permission from Ref. [257]. Copyright 2019 American Chemical Society.

## Data Availability

Not applicable.

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
