# Peer review of "Nanocomposite Hydrogels as Functional Extracellular Matrices"

_gels, 2023, doi:10.3390/gels9020153_

Round 1
Reviewer 1 Report
Some part of the manuscript has long detailed text. Thus, it is highly recommended that authors categorize these parts into subtitles so that authors can comprehend the text easily. Some notes are mentioned in order to make the manuscript more understandable. Please exert all notes below:
1. Some sentences have multiple references for example in page 1, line 31 and 33 and ....
Please reform this multiple references and put proper references at the end of each sentence
2.Some sentences of the manuscript doesn't have appropriate reference. For exame in page 1, line 21-23, the sentence " In living tissues ... migration, and polarization."
Please add proper reference at the end of each sentence of the manuscript (exept sentences that are well-established or the results of present panuscript)
3.In page 1, line 21-33, paragraph "In living tissues ... biodegradability, and electrical conductivity [6–8]. " authors have written about ECM and it's compositions but in the next paragraph (line 34-59), they have spoken about Tissue engineering wich has disrupted the continuity of the text. Please note that continuity of the text should be keeped.
4.In page 1, part" Hydrogels for tissue Engineering" . Authors have written this part of article in parts below:
1. some explanation about ECM and its compositions
2. some information about Tissue engineering
3. More data about ECM
4. Some information about Nano sensors and E-Field
5. Some scientific data about the role(s)of Nanoparticles and hydrogels in tissue engineering
6. The scope of the manuscript
7. More data of biomedical applications of nanoparticle-hydrogel
Mentioned order does not have continuity and proper structure of one section of a manuscript. Thus, the order of mentioned part should be:
1.ECM and it's features
2. The necessity, definition and the application of tissue engineering in the producing of artificial ECM
3. The applicetion of nanoparticles and their subgroups (especially hydrogels) in the field of manufacturing artificial ECM
4. The benefits and drawbacks of hydrogels for producing artificial ECM
5. Conclusion of the part " 1.Hydrogels for tissue Engineering"
Please reconsider the part " 1.Hydrogels for tissue Engineering " according to prior notes.
5.in page 4, line 127-133; the title of figure should contain brief and adequate information about the figure but not detailed information. (Detailed information should be discussed in the part" discussion" not the title of figure)
Please reconsider the title of Figue1
6.In page 4, line 135-142, the paragraph" In the absence of nano-additives... or reducing pore size." does not have proper reference in itself.
Note: please add suitable reference at the end of each sentence of your manuscript (exept well-established sentences and the results of the manuscript)
7.In page 4, line 134, the part" 2.1. Mechanical properties and anisotropy " can be written better. Please devide this part into
subgroups below:
2.1.1.Mechnical properties of Nano-reinforced tissue scaffolds
2.1.2. anisotropy of Nano-reinforced tissue scaffolds
After that, please deiscuss about The
1) conventional nanocomposite hydrogel systems (wich is discussed in line 150- 180) and
2)Nano-reinforced tissue scaffolds with magnetic features
Separately or in one of the mentioned subtitles so that other readers can read and understand the part "2.1. Mechanical properties and anisotropy " more easily.
8. Pleas write one or more sentences about the conclusion at the end of each part of the manuscript (this makes readers to conclude the informations which are provided in each part)
9.In page 8, please rewrite the part 2.2. Electrical conductivity" according to order below:
2.2.1.Electro-active nanocomposite scaffolds for the study of cardiac cell behavior
2.2.1.1.Electrical conductivity of carbon based NPs
2.2.1.2.Electrical conductivity of metal based NPs
2.2.3.Electro-active nanocomposite scaffolds for the study of neuronal cell behavior
After that, please discuss about each one and write about prior relevant scientific studies and their usages as tissue scaffold and also theor weakpoints
11. About the part" 3.1. Thermal stimulation"
Please exert the notes below:
1. Explain about other nanoparticles (which are relevant to hydrogels) which are used in other related studies
2. About GNPs: please write about the weakpoints of GNPs at the end of this part and compare GNPs with other nanocompositions
3. Rewrite this part according to order below:
First: the importance of thermal stimulation in in-vivo condition
Second: the ability to give proper response to thermal stimulations by nanoparticles
Third: write about different types of NPs (usages, weakpoints and other necessary informations) which have been studied in related studies
Forth: compare different types of NPs which have been studied in related studies with together
4.add conclusion at the end of this part based on the information you have discussed about
12. In page 18, line 716- 733
Please recosider the title of figure 6. It should
contain a brief information about figure and detailed data must be mentioned in the main text of the manuscript.
13. In page 22, line 897-906; please add proper reference at the end of sentences.
14.Please categorize the part" 4.2.1.Temperature" in three subtitle including:
4.2.1.1.dye-based temperature sensing
4.2.1.2.metal-organic-frameworks (MOFs)
4.2.1.3.Rare-earth ions temperature sensing
After that please discuss about their mechanisms, weakpoints and compare them with together based on prior scientific researches.
15.In the part" Conclusion", line 1047-1077
Please write about your conclusions and your suggestions based on your conclusions.(please omit more explanations and write this part more fluent with logical continuity and
based on the importance of your each conclusion)
16. Please check references carefully
Reviewer 2 Report
The review article "Nanocomposite hydrogels as functional extracellular matrices" written by Jooken et al. is a well-documented work on platforms that can be used as 3D extracellular matrices. It describes in detail the mechanical, anisotropic, and electrical properties of such ECMs. The different kinds of cell stimulation like thermal, electric, transduction, and mechanical can be applied to the ECMs. The different applications in sensing and monitoring biomarkers, temperature, mechanical changes, and electrical activity.
Author Response
No modifications requested
Reviewer 3 Report
In this review article, authors present an overview of the technological advances regarding the incorporation of functional nano-materials in artificial extracellular matrices, highlighting both passive and dynamically tunable nano-engineered components. The review article has been prepared quite comprehensively and will contribute to the literature.My contributions are below.
Can the effect of nanoparticles on viscoelastic properties be explained?
The effects of formulation parameters of nanoparticles can be evaluated.
Round 2
Reviewer 1 Report
Article in the current condition with the considered corrections can be published.